# *Tritrichomonas foetus* Cell Division Involves DNA Endoreplication and Multiple Fissions

Lucrecia S. Iriarte,[a,b] Cristian I. Martinez,[a,b] Natalia de Miguel,[a,b] Veronica M. Coceres[a,b]

[a]Laboratorio de Parásitos Anaerobios, Instituto Tecnológico Chascomús (INTECH), CONICET-UNSAM, Chascomús, Argentina
[b]Escuela de Bio y Nanotecnologías, Universidad Nacional de San Martin (UNSAM), Buenos Aires, Argentina

**ABSTRACT** *Tritrichomonas foetus* and *Trichomonas vaginalis* are extracellular flagellated parasites that inhabit animals and humans, respectively. Cell division is a crucial process in most living organisms that leads to the formation of 2 daughter cells from a single mother cell. It has been assumed that *T. vaginalis* and *T. foetus* modes of reproduction are exclusively by binary fission. However, here, we showed that multinuclearity is a phenomenon regularly observed in different *T. foetus* and *T. vaginalis* strains in standard culture conditions. Additionally, we revealed that nutritional depletion or nutritional deprivation led to different dormant phenotypes. Although multinucleated *T. foetus* are mostly observed during nutritional depletion, numerous cells with 1 larger nucleus have been observed under nutritional deprivation conditions. In both cases, when the standard culture media conditions are restored, the cytoplasm of these multinucleated cells separates, and numerous parasites are generated in a short period of time by the fission multiple. We also revealed that DNA endoreplication occurs both in large and multiple nuclei of parasites under nutritional deprivation and depletion conditions, suggesting an important function in stress nutritional situations. These results provide valuable data about the cell division process of these extracellular parasites.

**IMPORTANCE** Nowadays, it's known that *T. foetus* and *T. vaginalis* generate daughter cells by binary fission. Here, we report that both parasites are also capable of dividing by multiple fission under stress conditions. We also demonstrated, for the first time, that *T. foetus* can increase its DNA content per parasite without concluding the cytokinesis process (endoreplication) under stress conditions, which represents an efficient strategy for subsequent fast multiplication when the context becomes favorable. Additionally, we revealed the existence of novel dormant forms of resistance (multinucleated or mononucleated polyploid parasites), different than the previously described pseudocysts, that are formed under stress conditions. Thus, it is necessary to evaluate the role of these structures in the parasites' transmission in the future.

**KEYWORDS** *Tritrichomonas foetus*, cell division, endoreplication, multiple fission, fission multiple

The phylum Parabasalia comprises a monophyletic group of diverse species of flagellated protists, whose taxonomic classification has been based on morphological differences (particularly in the arrangement of the basal bodies of the flagella and associated cytoskeletal elements). Historically, the Parabasalia have been treated as 2 groups: the "trichomonads" (up to 6 flagella), and the "hypermastigotes" (multiflagellate) (1, 2). Trichomonads are flagellated protozoans that reside as parasites or commensals on animals and humans. *Tritrichomonas foetus* (from class Tritrichomonadea and order Tritrichomonadida) has a worldwide distribution, and is the causative agent of bovine tritrichomonosis, a venereal disease that causes early embryonic death, abortion, and infertility (3, 4). *T. foetus* also infects the ileum, cecum, and colon in domestic cats, and is known to be a primary cause of large bowel diarrhea in these

Address correspondence to Veronica M. Coceres, coceres@intech.gov.ar.

The authors declare no conflict of interest.

animals (5, 6). On the other hand, these protozoa have been described as commensal and facultative pathogens in pigs (7). *Trichomonas vaginalis* (from class Trichomonadea and order Trichomonadida) causes trichomoniasis, a common human sexually transmitted infection linked to pelvic inflammatory disease, adverse pregnancy outcomes, and an increased risk of HIV, papillomavirus infection, and cervical or prostate cancer (8–11).

Despite these serious health-related and economic consequences, biological processes, such as cell division, are poorly explored in these protozoans. Cell division is a crucial process in most living organisms that leads to the formation of 2 daughter cells from a single mother cell. Previous reports on the division process in *T. foetus* and *T. vaginalis* claimed that the mode of reproduction is exclusively by binary fission (12, 13). This process is characterized by the persistence of the nuclear membrane and asymmetric extranuclear spindle (closed extranuclear pleuromitosis) (12, 14). Furthermore, it has been shown that skeletal structures such as axostyles and flagella are engaged in karyokinesis and cytokinesis, respectively (15).

Although fission binary is the most common cell division mode identified in many cell types, multinuclearity has also been described across the living world. However, the mechanisms by which it occurs, and the potential benefits connected with it, are still unclear in most cases. Multinucleation refers to the number of physically separated nuclei within a common cytoplasm, which is a result of: (i) cell-cell fusion (syncytium); or (ii) multiple rounds of nuclear division within a common cytoplasm (coenocyte) (16, 17). It is known that multinucleated cells are important in processes as diverse as fruit fly early development, bone remodeling (18), placenta formation (19), and cancer metastasis (20). Cells with multiple nuclei have been described in plants, fungi, and protozoans, and have been associated with possible functional advantages, especially in adverse environmental conditions. In Amoebozoa, multinuclearity is found in many species (*Entamoeba histolytica*, *Naegleria gruberi*, and *Hartmannella rhysodes*), and despite being an integral part of their life cycle, there is a lack of evidence concerning its potentially associated functions (21–23). Furthermore, it has been reported that *Plasmodium falciparum* nuclei divide asynchronously during some stages of infection, forming multinucleate cells before egress from the host cell (24). Specifically, in trichomonads, there is scarce information reported to the occurrence of multinucleate forms in *Trichomonas vaginalis*, *Trichomonas augusta*, *Trichomonas lacerate*, *Trichomonas muris*, *Eutrichomonas serpentis*, and *Tetratrichomonas prowazeki* (25, 26). Specifically, in *Tritrichomonas foetus*, multinucleate cells have only been related to pseudocystic structures to date (27, 28). Pseudocysts have been described as trophozoites that adopt a spherical shape and internalize their flagella in unfavorable environmental conditions, but their role in the trichomadidae life cycle is still unknown (29).

According to the bibliography, multinucleate forms are closely related to posterior cell division by multiple fission in other trichomonads (25). While binary fission is the most common mode of cell division, some algae, protozoans, and true slime molds (Myxomycetes) commonly divide by multiple fission. The nucleus undergoes numerous mitotic divisions in such instances, resulting in a large number of nuclei. The cytoplasm splits when nuclear divisions are completed, and each nucleus is enclosed in its own membrane to form a unique cell. Interestingly, it has been reported that this type of fission parental body occurs in some species in unfavorable environmental conditions. In this context, it has been described that multiple fission occurs in *Trichomonas augusta*, and other trichomonad flagellates, as a normal phase of the life cycle (25). Alternatively, multinuclearity could be originated by fusion events between uninucleate cells or by the capacity of some cells to replicate their DNA content without concluding the cytokinesis process (30, 31). Cellular endoreplication is a cell cycle variant in which the genome is re-replicated without mitosis, resulting in cellular polyploidization before segregation rather than the conventional simple duplication. It was also shown that certain cell types respond with endoreplication cycles before an exogenous stimulus. As examples, *Entamoeba histolytica* has endoreplication cycles that result in cells with polyploid nuclei or multinucleate cells in different growth conditions in media culture (32).

Here, we show that multinuclearity is a phenomenon regularly observed in different

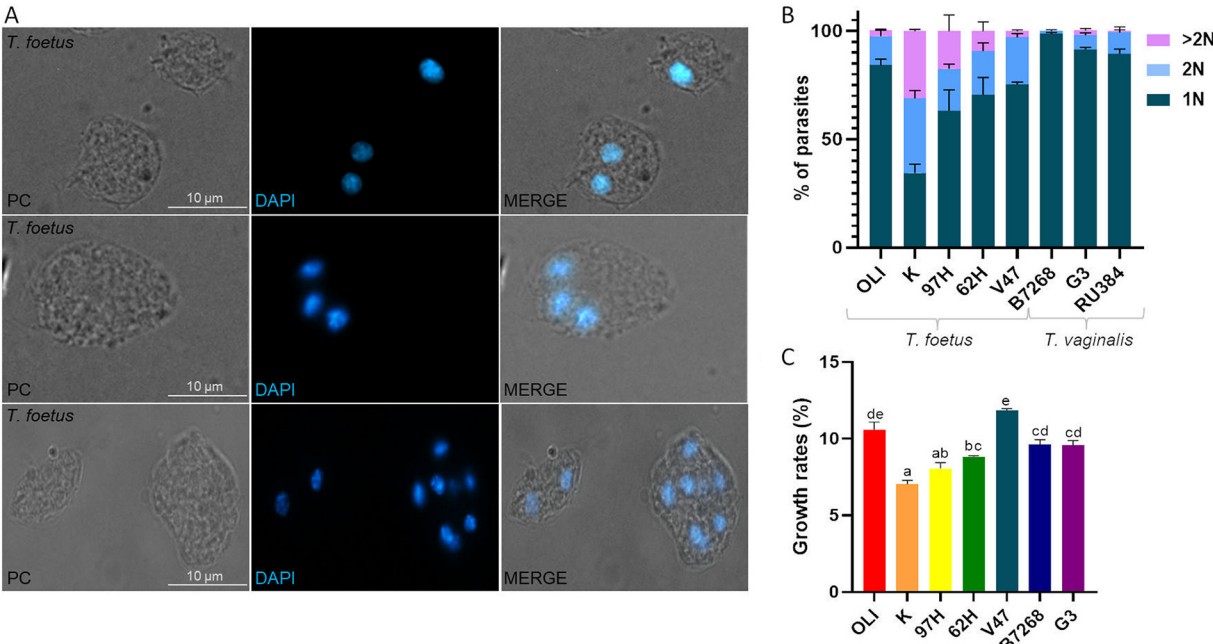

**FIG 1** Nuclei number heterogeneity in *T. foetus* and *T. vaginalis* under standard growth conditions. (A) Representative images of *T. foetus* parasites with 1, 2, or more than 2 nuclei per cell. Nuclei were stained with DAPI (blue). PC, phase-contrast. Scale bar, 10 $\mu$m. (B) Quantification of the number of nuclei per parasite. One hundred parasites from each population were counted in triplicate in 3 independent experiments. The percentages of parasites with 1, 2 or more than 2 nuclei are shown. Different *T. foetus* (OLI, K, 97H, V47, and 62H) and *T. vaginalis* (B7268, G3, RU384, and CDC1132) strains were analyzed. The results represent the average of 3 independent experiments and the standard error (SE). (C) Growth rates of *T. foetus* (OLI, K, 97H, 62H, and V47) and *T. vaginalis* (B7268 and G3) strains. The average of 3 independent experiments is represented by the bars, and the standard deviation is represented by the error bars. Mean differences were calculated using Tukey's test with an alpha = 0, 05. The rates are calculated from the concentration of parasites in the stationary phase and the time it takes to reach that phase.

*T. foetus* and *T. vaginalis* strains. Although multinuclearity could be observed in standard culture conditions, it clearly increased upon nutritional stress. Importantly, when the standard culture media conditions are restored, the cytoplasm of these multinucleated cells separates, and numerous parasites are generated in a short period of time by the fission multiple, suggesting that multinuclearity is associated to a dormant stage (different to the previously described pseudocyst stage).

Additionally, we demonstrated that nutritional depletion or nutritional deprivation led to different dormant phenotypes. Although multinucleated *T. foetus* were mostly observed during nutritional depletion, numerous cells with 1 larger nucleus has been observed under nutritional deprivation conditions. We also revealed that DNA endoreplication occurs in both large and multiple nuclei of parasites under nutritional deprivation and depletion conditions, suggesting an important function in stress nutritional situations. Taken together, our findings contribute to our understanding of *T. vaginalis* and *T. foetus* cell division, and provide the basis for further research in this area.

## RESULTS

**Variations in nuclear number are inherent features of *T. vaginalis* and *T. foetus* parasites.** It is widely accepted that *T. vaginalis* and *T. foetus* divide by longitudinal binary fission, where a parent cell splits into 2 identical mononucleate daughter cells (12, 13). However, we usually observed that *T. foetus* (K strain) contained multiple nuclei in some parasites under standard growth conditions (Fig. 1A). Based on these frequent observations, we assessed the number of nuclei in different *T. foetus* and *T. vaginalis* strains using DAPI nuclear staining and fluorescence microscopy. Analysis of parasites in the logarithmic growth phase demonstrated that *T. foetus* contained greater heterogeneity in the number of nuclei per cell compared to *T. vaginalis* strains (Fig. 1B). The *T. foetus* K strain, in particular, had the highest variance in the number of nuclei per

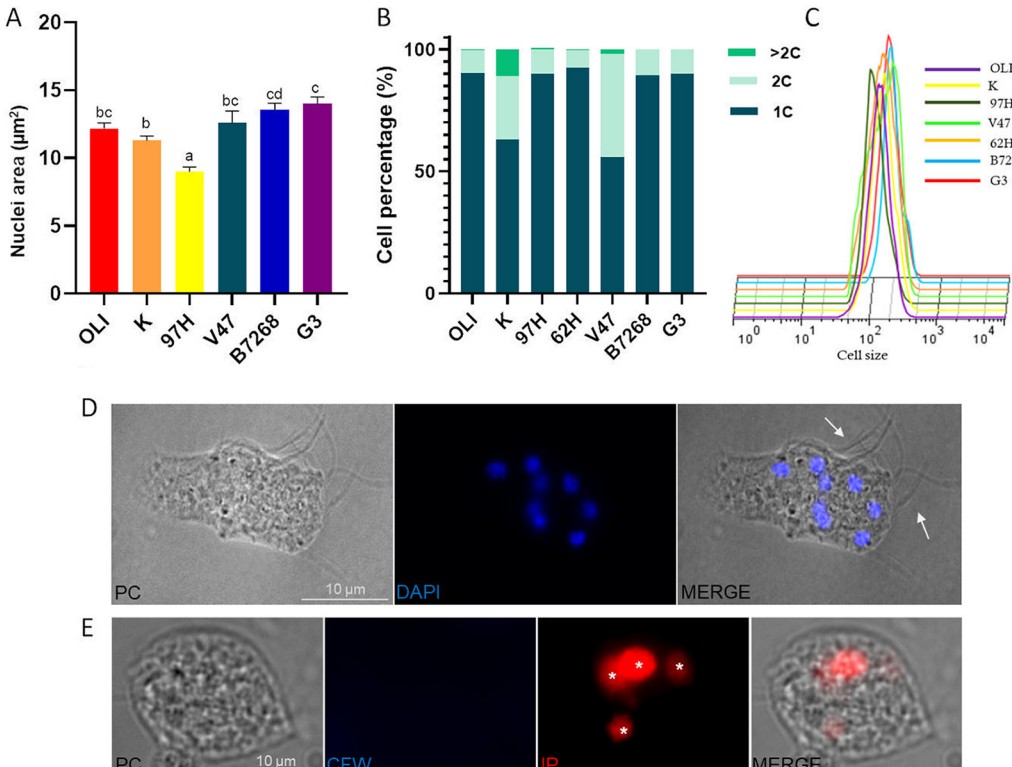

**FIG 2** Multinucleated parasites are present in standard culture conditions. (A) Estimation of the nuclear size of *T. foetus* (OLI, K, 97H, and V47 strains) and *T. vaginalis* (B7268 and G3 strains). The images of 50 nuclei staining with DAPI were analyzed using Image J's threshold tool and the "Analyze Particles" function. The experiments were carried out in triplicate and three times. Mean differences were calculated using Kruskal-Wallis test with a *P*-value of 0,05. (B) The DNA content of *T. foetus* (OLI, K, 97H, 62H, and V47 strains) and *T. vaginalis* (B7268 and G3 strains) was determined by flow cytometry. (C) Flow cytometric analysis of cell size of *T. foetus* K strain, *T. foetus* OLI, K, 97H, and V47 and 62H strains, and *T. vaginalis* B7268 and G3 strains. (D) Representative image of *T. foetus* K multinucleate with external flagella. PC, phase-contrast image. DAPI (blue). The white arrows indicate the external flagella. Scale bar, 10 $\mu$m. (E) *T. foetus* with more than 2 nuclei were stained with Calcofluor white (blue). White asterisks indicate nuclei stained with propidium iodide (red). Scale bar, 10 $\mu$m.

parasite, with 34% of the cells having 2 nuclei, and 35% of the parasites having more than 2 nuclei. Additionally, *T. foetus* 97H and 62H strains contained a high percentage of parasites with 2 or more than 2 nuclei (Fig. 1B). Despite the differences observed in the number of nuclei, we observed that *T. foetus* K and 97H showed significantly lower growth when the growth rates were calculated for the strains analyzed (Fig. 1C). These results might be suggesting that the number of nuclei would not be related to a greater efficiency in parasite multiplication under standard growth culture conditions.

The size of the nuclei of different strains was then analyzed. The obtained results indicate that the nuclei of different strains had different sizes (Fig. 2A). Within the strains of *T. foetus*, the sizes varied between 8.99 $\mu$m$^2$ and 12.58 $\mu$m$^2$, corresponding to strains 97H and V47, respectively. In the *T. vaginalis* strains, the nuclei sizes ranged between 13.56 $\mu$m$^2$ (B7268 strain) and 16.56 $\mu$m$^2$ (RU384 strain). Based on the observed nuclei size differences, we subsequently evaluated whether *T. foetus* and *T. vaginalis* strains exhibited variation in their DNA content. To this end, the parasites were fixed, the DNA was stained with propidium iodide (PI), and the DNA content was measured by flow cytometry. We observed that *T. foetus* DNA content was heterogeneous: OLI strain showed 90.19%, 9.55%, and 0.25% cells with 1C, 2C, and > 2C nuclear DNA content, respectively. In *T. foetus* K parasites, we observed that 63%, 26.13%, and 10.95% of the cells showed 1C, 2C, and > 2C nuclear DNA content, respectively. V47 strain presented a 42% of cells with 2C DNA content. Likely, the difference between the cell count of 1 or 2 nuclei, and cells with twice the DNA content could be to the presence of parasites with nuclei with more than 1C DNA content. While *T.*

*vaginalis* showed greater homogeneity in DNA content, in B7268 and G3 strains, most of the parasites (90%) presented 1C nuclear DNA content (Fig. 2B). Considering that polyploidy is associated with a larger cell size in bacteria, plants, and mammals (33–35), we evaluated the cell size of *T. foetus* and *T. vaginalis* strains by flow cytometry. In Fig. 2C, we showed that, despite the DNA heterogenicity observed in some cases, the different strains of *T. foetus* and *T. vaginalis* did not show variation in their cell size.

The trophozoites (a polar and flagellated cell) can adopt a spherical shape with internalized flagella, known as "pseudocystic" under unfavorable environmental conditions (29). This pseudocystic form can carry out nuclear division, and form multinucleated cells (27, 29). Based on these antecedents, we evaluated whether the *T. foetus* multinucleated cells observed in culture media conditions were pseudocysts or not. As observed in Fig. 2D, some *T. foetus* multinucleated cells have external flagella (Fig. 2D). To confirm these results, *T. foetus* parasites grown in logarithmic phase were fixed and stained with Calcofluor White Stain (CFW), a nonspecific fluorochrome that binds to the pseudocysts/cysts surface (36). In concordance with our previous results, numerous multinucleate parasites were not "pseudocystic forms" (Fig. 2E). Our results demonstrate the presence of multinucleated parasites that are not pseudocysts in standard culture conditions in *T. foetus*.

**Nutrient depletion induces the formation of multinucleate *T. foetus* parasites.** To analyze the presence of multinucleated parasites and/or pseudocysts under stress conditions, we evaluated the impact of nutrient starvation in the culture media. We compared parasite strains that contained different nuclei numbers under standard culture conditions. *T. foetus* parasites (K and 62H strains) were synchronized by 12 h incubation in a media without serum (37). Then, the serum was added to the media, and the parasites growth was monitored each 12 h until their death phase. The samples of each time point were fixed, the nuclei were stained with DAPI, and quantified by fluorescence microscopy. In Fig. 3A we showed the presence of *T. foetus* K multinucleate parasites at 36 h under these incubation conditions.

As shown in Fig. 3B, in unsynchronized *T. foetus* K parasite culture (UP), we found 18.7% and 6% of parasites with 4 nuclei (4N) and more than 4 nuclei (4N), respectively. In synchronized parasites (SP), we observed a 5.7% (4N) and a 0.3% (>4N); suggesting that the lack of nutrients enriched a population of parasites with 1 nucleus. At 24 h, 18% of *T. foetus* K parasites had 4N, and 2.4% had > 4N. At 36 h, the results were: 27.3% (4N) and 5.6% (>4N). *T. foetus* 62H strain showed 5.8% and 0.9% of parasites with 4N and > 4N, respectively at 24 h, and 0.3% of parasites with 4N at 36 h. Based on these results, we concluded that nutrient depletion induces the formation of multinucleate parasites, mainly in the *T. foetus* K strain.

Afterwards, we compared the total DNA content per parasite in 2 different nutrient depletion conditions (parasites grown in the absence of serum for 36 h and parasites grown for 36 h without change of the media culture). *T. foetus* K produced 34.15% and 50.2% of parasites with more than twice the DNA content (>2C), respectively, in parasites grown for 36 h without changing the media culture, and parasites grown in the absence of serum for 36 h. These results were consistent with previous results where, under unfavorable conditions, *T. foetus* K formed multinucleate structures. *T. foetus* 62H strain had 25.54% and 47.57% of parasites with more than twice the DNA content (>2C) in parasites grown for 36 h without changing the media culture, and parasites grown in the absence of serum for 36 h, respectively (Fig. 3C). These findings suggest that, whereas the number of nuclei varies under nutritional stress, the DNA content of different *T. foetus* strains does not show significant differences. Then, the size of the nuclei of *T. foetus* K and *T. foetus* 62H strains grown for 36 h without changing the media culture, and parasites grown in the absence of serum for 36 h was analyzed. The obtained results indicate that the average nuclei size was 50.54 $\mu$m² and 34.35 $\mu$m² for *T. foetus* K parasites grown for 36 h without changing the media culture (horse serum [HS]+) for (HS+), and parasites grown in the absence of serum for 36 h (HS−), respectively. In the *T. foetus* 62H strain, the average nuclei size was 42.65 $\mu$m² and 70.35 $\mu$m² for parasites grown for 36 h without changing the media culture (HS+), and parasites grown in the absence of serum for 36 h (HS−), respectively.

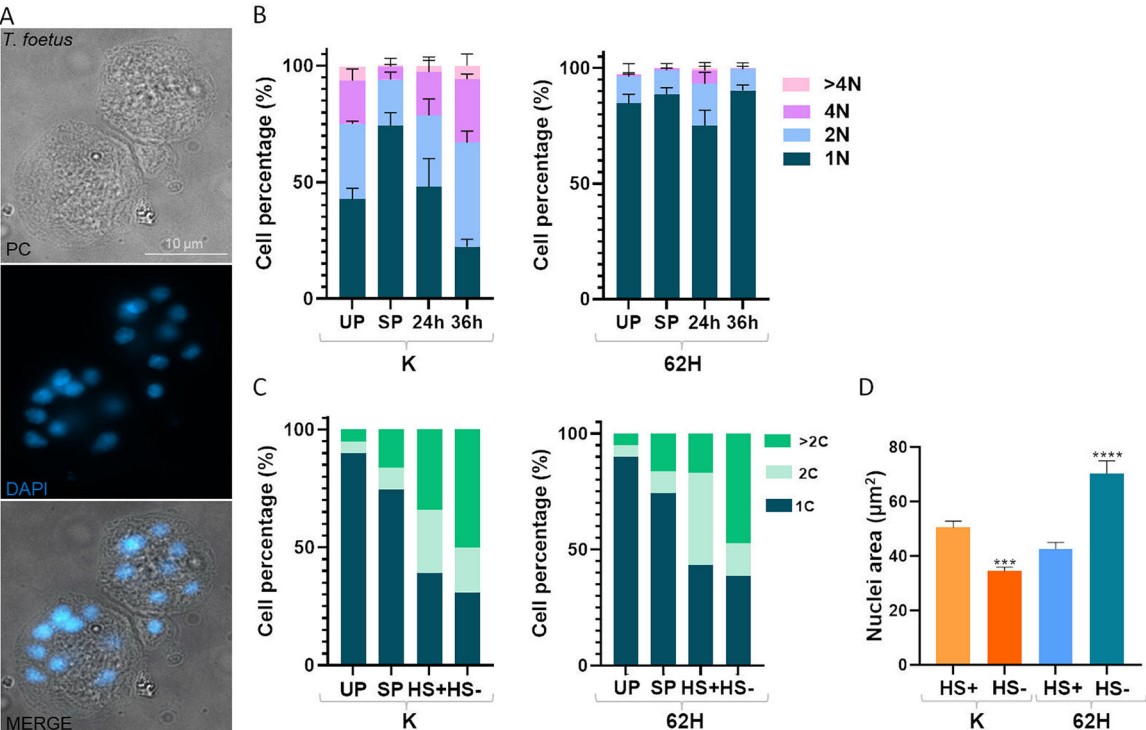

**FIG 3** Analysis of the formation of multinucleated parasites under conditions of nutrient depletion. (A) Representative image of *T. foetus* K (multinucleate) grown in the presence of serum (without culture media change) for 36 h, stained with DAPI (blue), and visualized by epifluorescence microscopy. PC, phase-contrast image. Scale bar, 10 $\mu$m. (B) Quantification of the number of nuclei per parasite stained with DAPI of *T. foetus* strains K and 62H, was conducted after they were grown in the presence of serum (without culture media change) until their death phase. UP, unsynchronized parasites' culture. SP, synchronized parasites' culture. Samples were taken at 24 and 36 h. The percentages of parasites of different *T. foetus* strains with 1, 2, 4, or more than 4 nuclei are shown. One hundred parasites from each population were counted in triplicate in 3 independent experiments. The results represent the average of 3 independent experiments, and the standard error (SE). (C) The DNA content profile of asynchronous (UP) and synchronized parasites (SP) grown for 36 h in the absence (HS-) or presence (HS+) of horse serum was determined by flow cytometry. (D) Estimation of the nuclear size of *T. foetus* (K and 62H strains) grown for 36 h in the absence (HS-) or presence (HS+) of horse serum. The images of 50 nuclei staining with DAPI were analyzed using the Image J's threshold tool and the "Analyze Particles" function. Experiments were performed in triplicate and repeated three times. Mean differences were calculated using the Kruskal-Wallis test with a *P*-value of 0.05.

Interestingly, we concluded that nutrient depletion does not induce the formation of multinucleated parasites but induces an increase in the size of the nuclei in the *T. foetus* 62H strain (Fig. 3D).

As multinucleated cells were observed upon nutrient depletion, we next determined if parasites grown for 36 h without changing the culture media were pseudocysts. To this end, *T. foetus K* parasites were stained with CFW (Fig. 4A), and analyzed by fluorescence microscopy. We could observe that 13.75% of *T. foetus* K parasites analyzed were pseudocysts in these conditions of growth (Fig. 4B). Thus, not all multinucleated parasites under conditions of nutritional stress during 36 h were pseudocysts.

Trophozoite and pseudocyst forms of *T. foetus* and *T. vaginalis* parasites contain a cytoskeleton organized into highly differentiated structures, including the pelta-axostyle, costa, and flagella, which are already duplicated before the beginning of mitosis (S/G2 phase) (15, 27). Hence, we analyzed the cytoskeletal structure distribution in multinucleated parasites obtained after 36 h without changing the growth media by indirect immunofluorescence (IFA), using an anti-alpha tubulin antibody and DAPI for nuclei staining. In concordance with previous reports, we detected a cytoskeleton (axostyle and flagella) per nuclei in these multinucleate parasites (Fig. 4C).

**A multiple fission process occurs during *T. foetus* and *T. vaginalis* cell division.** Some organisms (e.g., algae and some protozoans) regularly divide by multiple fission (38, 39). In such cases, the nucleus undergoes several mitotic divisions, generating a

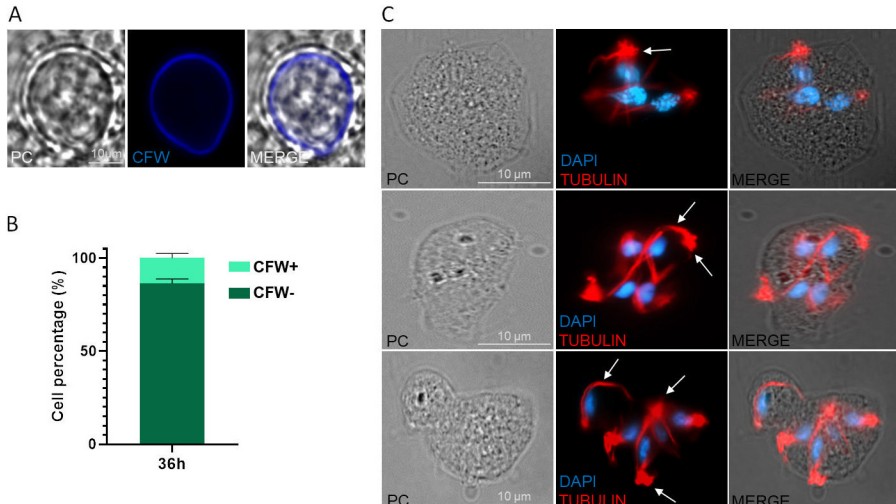

**FIG 4** Evaluation of the presence of pseudocysts and the cytoskeletal structure distribution in multinucleate parasites induced by nutrient depletion conditions. (A) A representative image of a pseudocyst stained with Calcofluor white (CFW) in a *T. foetus* K grown *in vitro* for 36 h without changing the culture media. The samples were analyzed by epifluorescence microscopy. Scale bar, 10 $\mu$m. (B) Graph representing the percentage of parasites CFW+ and CFW- in samples of *T. foetus* K grown *in vitro* for 36 h without changing the culture media. One hundred parasites were counted in triplicate in 3 independent experiments. (C) The axostyle and flagella of *T. foetus* parasites grown without change of medium for 36 h were labeled with mouse anti-tubulin antibody (red) and DAPI (blue). PC, phase-contrast image. The white arrows indicate a cytoskeletal structure (axostyle and flagella) related to each nucleus. Scale bar, 10 $\mu$m.

higher number of nuclei per cell. Then, the cytoplasm separates to form numerous cells with only a nucleus. Interestingly, this type of parental cell fission has been described in some organisms only when they are under unfavorable conditions (40). In this sense, multiple fission occurs in other trichomonad flagellates as a normal phase of the life cycle, in which this process results in the formation success of 2 to 4, and 4 to 8 nuclei with the accompanying multiplication of extranuclear organelles, such as motor apparatus (including undulating membrane, flagella, and the axostyle) (25).

Here, we showed that the number of *T. foetus* multinucleate parasites increased under unfavorable nutritional conditions. To evaluate if these multinucleate cells could produce several individual parasites when the media is restored, we transferred the *T. foetus* K parasites grown in the absence of nutrients (complete culture medium for 36 h or serum-free medium for 36 h) into fresh media and estimated growth rates. The parasites were counted at 36 h and 4 h after the changes in media, and we calculated the growth rate (Fig. 5A). The results showed that the growth rate was greater in parasites grown in complete culture medium for 36 h. Parasites cultured in a serum-free medium for 36 h (under severe stress conditions) were unable to complete cell division with the same efficiency.

To examine the cytoskeletal structures of multinucleated parasites during the egress, parasites *T. foetus* K and *T. vaginalis* CDC1132 strain over-expressing TvTSP6-HA, a protein strongly localized in the parasite flagella (41), were grown in the same culture media for 36 h. Then, the parasites were placed in fresh media culture and incubated for 2 more hours. Later, the parasites were fixed, and an IFA was performed using an anti-alpha tubulin antibody and DAPI for nuclei staining (*T. foetus*); an anti-HA antibody, anti-alpha tubulin antibody, and DAPI for TvTSP6-HA parasites (*T. vaginalis*).

In *T. foetus* (Fig. 5B) and *T. vaginalis* (Fig. 5C), it can be observed that each nucleus from multinucleated parasites is paired with cytoskeletal structures (flagella and axostyle) when the fresh, complete media is restored. Furthermore, it can be observed that cell division can occur in any of 3 planes: longitudinal, transverse, or oblique (Fig. 5B and C, and Fig. 6A). On the other hand, when the segregation of nuclei and/or its coordination with cytokinesis appear to be compromised in other organisms, it has been

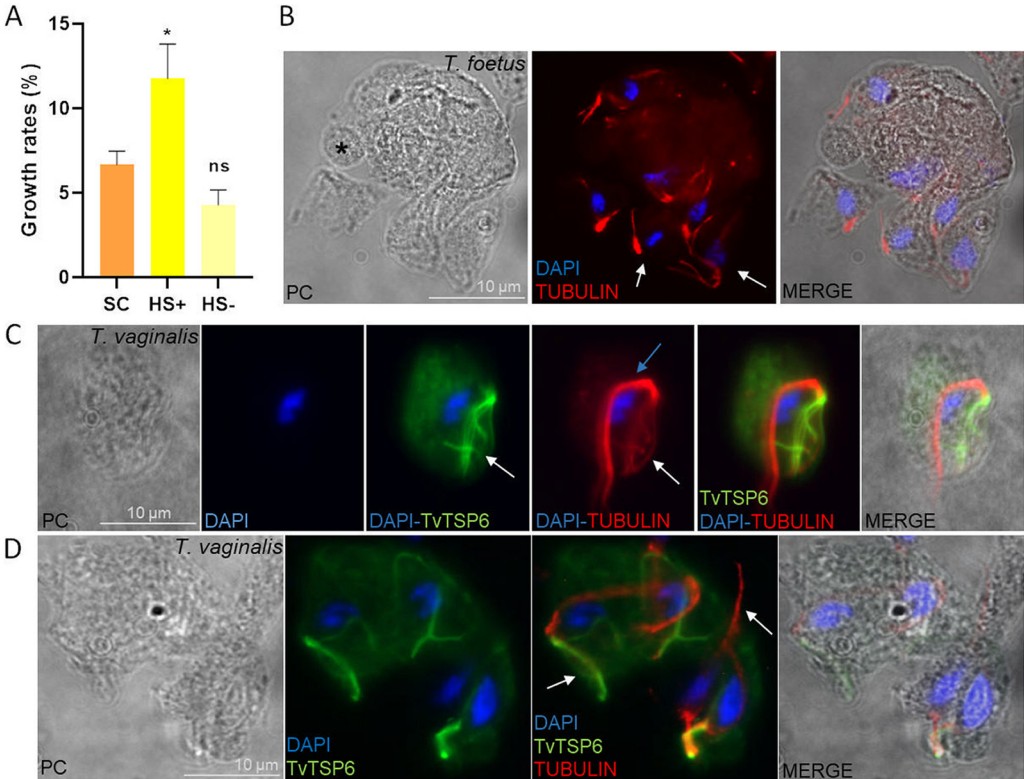

**FIG 5** Examination of the process of budding from *T. foetus* multinucleated parasites. (A) Growth rates of parasites grown under standard conditions (SC), in the same complete culture medium (HS+) for 36 h, and in a serum-free medium (HS-) for 36 h were compared. The parasites transferred into fresh growth were counted at 0 h and 4 h after the changes in media. Error bars represent standard error, and asterisks denote statistically significant differences determined by Welch's test. (B) The axostyle and flagella of *T. foetus* (maintained in the same complete culture medium HS+ for 36 h) during the budding process was labeled with mouse anti-tubulin antibody (red) and DAPI (blue) and analyzed using PC, phase-contrast image. The white arrows indicate a cytoskeletal structure related to each nucleus. The black asterisk indicates a "zoid". Scale bar, 10 μm. (C) Indirect immunofluorescence of *T. vaginalis* parasites overexpressing the TvTSP6 flagella protein with a C-terminal hemagglutinin (HA) were stained in standard growth conditions using a rabbit anti-HA antibody (green). The axostyle and flagella were labeled with a mouse anti-tubulin antibody (red), and the nuclei were stained with DAPI (blue), and analyzed through PC, phase-contrast image. The white arrows indicate flagella, and the blue arrows indicate the axostyle (a cytoskeletal structure related to the nucleus). Scale bar, 10 μm. (D) Images show multinuclear *T. vaginalis* parasites overexpressing the TvTSP6 flagella protein with a C-terminal hemagglutinin (HA) grown without change of medium for 36 h. Indirect immunofluorescence was carried out using a rabbit anti-HA antibody (green). The cytoskeleton was labeled with a mouse anti-tubulin antibody (red), and the nuclei were stained with DAPI (blue), and analyzed through PC, phase-contrast image. The white arrows indicate a cytoskeletal structure (axostyle and flagella) related to each nucleus. Scale bar, 10 μm.

linked to the production of enucleated daughter cells, known as zoids (42, 43). Here, we also observed the presence of enucleated parasites in *T. foetus* and *T. vaginalis*, which suggests that unequal division of multinucleate cells could produce zoids in trichomonads (Fig. 6B and C). The zoids were also demonstrated to not contain the classical cytoskeletal structures in these parasites (Fig. 5B and Fig. 6C).

In summary, *T. foetus* K strain and *T. vaginalis* CDC1132 strain form multinucleated cells in depleted nutrient conditions and that this process is reversible when the standard cultivation conditions are restored. Whereby, these dormant multinucleated forms could be used for these parasites as a survival and multiplication strategy under stress conditions.

**Nutrient deprivation induces the formation of spherical parasites with a single, and larger nucleus.** We next evaluated the presence of multinucleated parasites under extremely low nutrient levels. To this end, *T. foetus* K parasites were incubated in water for 24 and 48 h. Afterward, the number of nuclei per cell was quantified using DAPI staining. The incubation in water induced the transformation of pyrifom trophozoites into spherical shapes (Fig. 7A) with internalized flagella (Fig. 7B). Then, we determined the number of nuclei per parasite at 24 and 48 h of incubation in water. At 24 h,

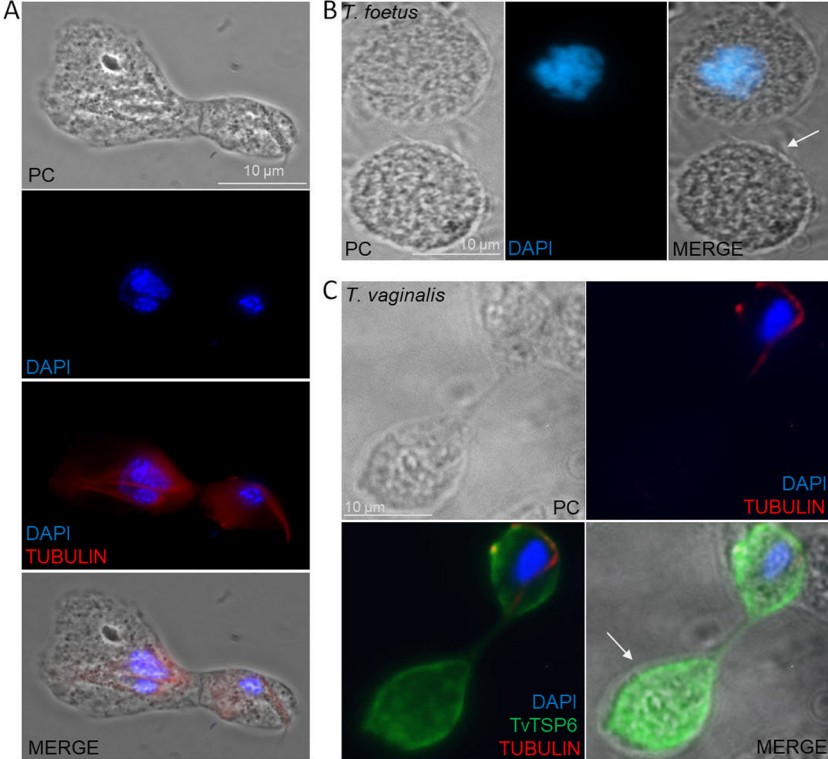

**FIG 6** Exploration of the cytokinesis process in the *T. foetus* parasite under nutrient depletion conditions. (A) The indirect immunofluorescence image shows that the longitudinal binary fission is not the only type of cell fission present in *T. foetus*. This image also demonstrates that the asynchronous nuclear division cycle occurs in these parasites. The cytoskeleton was labeled with a mouse anti-tubulin antibody (red), and the nuclei were stained with DAPI (blue), and analyzed through PC, phase-contrast image. Scale bar, 10 μm. (B) Representative image of *T. foetus* 62H enucleated daughter cells (zoids). Nuclei were stained with DAPI (blue) and analyzed through PC, phase-contrast. Scale bar, 10 μm. (C) Indirect immunofluorescence shows enucleated daughter cells (zoids) in TvTSP6 parasites during cytokinesis. The cytoskeleton was labeled with a mouse anti-tubulin antibody (red), TvTSP6 was labeled with an anti-HA antibody (green), and the nuclei were stained with DAPI (blue). The white arrows indicate the zoids in both parasites. Scale bar, 10 μm.

54.6%, 38.3%, and 6.3% of parasites contained 1, 2, and > 2 nuclei per parasite, respectively. At 48 h, the percentages of parasites with 1, 2, and > 2 nuclei were 76.6%, 21.3%, and 3.6%, respectively (Fig. 7C). Interestingly, when the parasites have just 1 nucleus, it has a larger size compared to control parasites (Fig. 7D and E).

To evaluate if these rounded cells with internalized flagella were pseudocyst, the parasites were stained with CFW. Only 8.6% and 12.6% of parasites kept for 24 and 48 h in water, respectively, were CFW positive (Fig. 7F).

Lastly, the DNA content of parasites incubated in water (W) was analyzed using propidium iodide (PI) staining by flow cytometry. As control, the DNA content of *T. foetus* K parasites grown for 24 h in standard culture media (SCM) was measured. As can be seen in Fig. 7G, 76% of the parasites incubated in water contain more than twice their DNA content. To evaluate if the parasites maintained in the water were able to convert to trophozoite and multiply in culture media, *T. foetus* K parasites incubated in the water for 24 h were transferred to Diamond's medium and incubated for 24, 48, and 72 h (Fig. 7H). After 24 h in SCM medium, the culture was composed of 71% pyriform motile trophozoites (PT), and 29% spherical-shaped trophozoites (ST). At 48 h, the percentages of pyriform trophozoites and spherical-shaped trophozoites in the culture were 93% and 7%, respectively. A total of 100% of pyriform trophozoites were found after the cells remained 72 h in culture media (Fig. 7H). We also observed that a single nucleus of spherical dormant parasites (Fig. 8A) divides into several "smaller nuclei" 3 h after nutrient supply (Fig. 8B). These results indicate that spherical dormant parasites

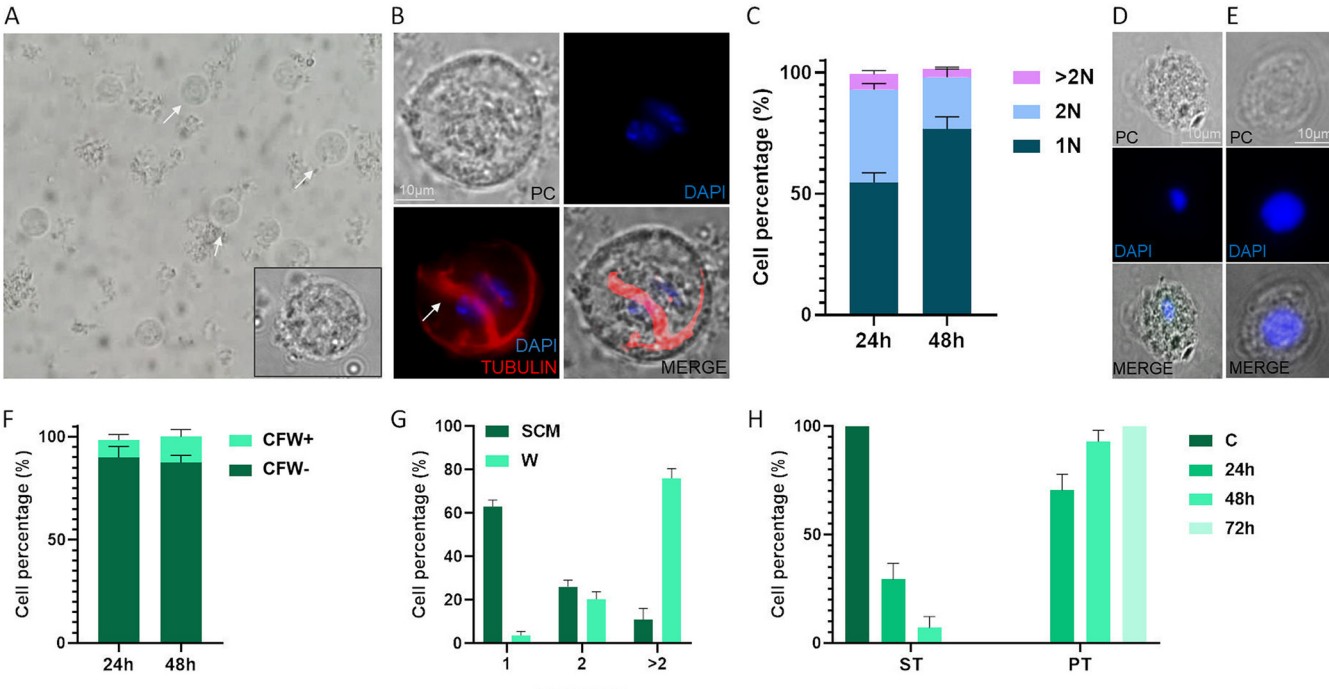

**FIG 7** Analysis of the *T. foetus* nuclei under conditions of nutrient deprivation. (A) A representative phase-contrast image of *T. foetus* after 24 h of incubation in water. The white arrows indicate trophozoite spherical shapes. (B) Indirect immunofluorescence using an anti-alpha tubulin antibody shows spherical parasites with internalized flagella (white arrow). DAPI (blue). PC, phase-contrast image. Scale bar, 10 $\mu$m. (C) Graph showing the number of parasite nuclei incubated in water for 24 and 48 h. (1N, one nucleus); 2N, two nuclei; >2N, three or more nuclei. (D to E) A representative images of *T. foetus* grown in culture media (control) and parasites *T. foetus* incubated in water during 48 h (with a single larger nucleus). Scale bar, 10 $\mu$m. (F) A graph showing the percentage of CFW+ and CFW- parasites in *T. foetus* K samples incubated in water for 24 and 48 h. One hundred parasites were counted in triplicate in 3 independent experiments. (G) A graph representing the DNA content profile of *T. foetus* K grown in standard culture media conditions (SCM) for 24 h, and incubated in water (W) for 24 h. DNA content was measured by flow cytometry. (H) Graph depicting the percentage of trophozoite spherical shapes (ST) and trophozoite pyriform shapes (PT) when *T. foetus* K parasites grown in the water for 24 h were recovered in Diamond's medium for 24 h, 48 h, and 72 h. (C) *T. foetus* K incubated in water as a control. (C, F, G, and H) The averages of each medium and their standard errors.

that form under nutrient starvation, with a single nucleus and more than 1C DNA content, can revert to motile trophozoites when the nutrient supply is restored.

**Genome endoreplication occurs during the cell cycle of *T. foetus*.** The endoreplication cycle is one common cell cycle variant in which cells increase their genomic DNA content without dividing. It is known that endoreplication is an intrinsic characteristic of *E. histolytica*, and in this organism, the formation of multinucleate cells is favored by the absence of nutrients (32, 44). Due to these antecedents, we evaluated whether cells with more than 1 nucleus or with larger nuclei continued to replicate their DNA in *T. foetus*. For this, we carried out an indirect immunofluorescence using an antibody that specifically recognize the "proliferating cell nuclear antigen" (anti-PCNA) to determine the active DNA replication in parasites under stress conditions (45, 46). We detected PCNA-positive *T. foetus* when parasites are grown for 36 h in the same media culture (Fig. 9A), indicating that, under conditions of low nutrients condition, the replication machinery is active in these parasites. To validate these results, we performed an assay using bromodeoxyuridine (BrdU), a thymidine analogue that is exogenously administered and incorporated into duplicating cells. *T. foetus* serum-starved parasites were incubated with BrdU for 2 h, and an indirect immunofluorescence using an anti-BrdU antibody was performed. In concordance with our previous results, we detected BrdU-positive nuclei (Fig. 9B), indicating that the DNA is actively synthesized, a requisite for the endoreplication process. As endoreplication is induced by colchicine (47), we then treated the parasites with 1.5 mM colchicine for 18 h and measured the DNA content by flow cytometry. As can be seen in Fig. 9C, the parasites treated showed 16.5%, 44.5%, and 38.5% of cells with 1C, 2C, and > 2C of DNA content, respectively. While in parasites not treated, the percentages were 83%, 10%, and

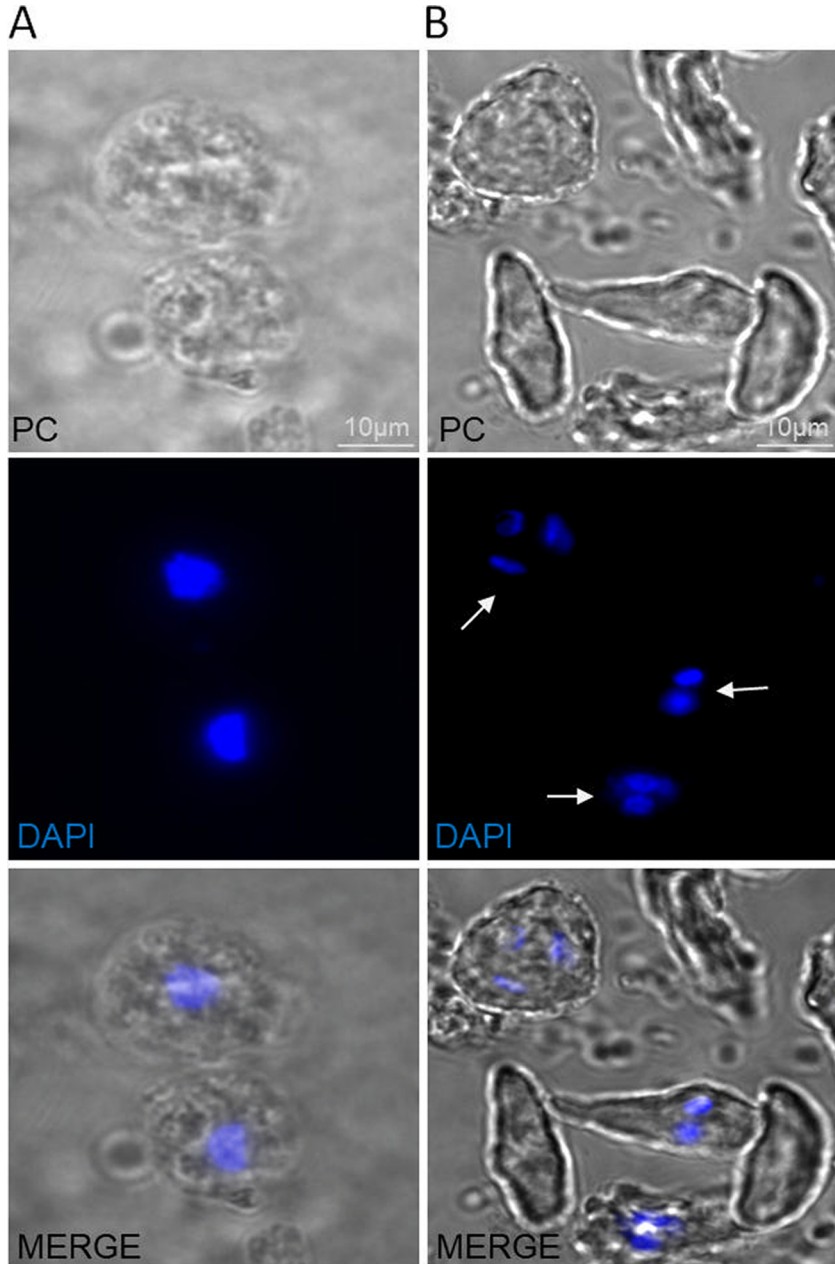

**FIG 8** Comparison of *T. foetus* nuclei under conditions of nutrient deprivation, and after nutrient addition. (A) Representative image of *T. foetus* parasites incubated in water for 24 h (spherical trophozoites). Scale bar, 10 $\mu$m. (B) Representative image of *T. foetus* parasites incubated in water for 24 h and recovered in Diamond's medium for 4 h (pyriform trophozoites are more abundant). Nuclei were stained with DAPI (blue). PC, phase-contrast. The white arrows indicate the presence of numerous small nuclei. Scale bar, 10 $\mu$m.

7% for 1C, 2C, and > 2C of DNA content, respectively. Also, by nuclei staining (DAPI), we detected the larger nuclei's presence (Fig. 9D and E). As a result, we concluded that these parasites are able to actively replicate their DNA under stress conditions by an endoreplication process.

## DISCUSSION

Like most living organisms, in trichomonads, cell division was widely considered a central process where 2 daughter mononucleated cells are generated from 1 mother cell. However, we demonstrated that different *T. foetus* and *T. vaginalis* strains possess

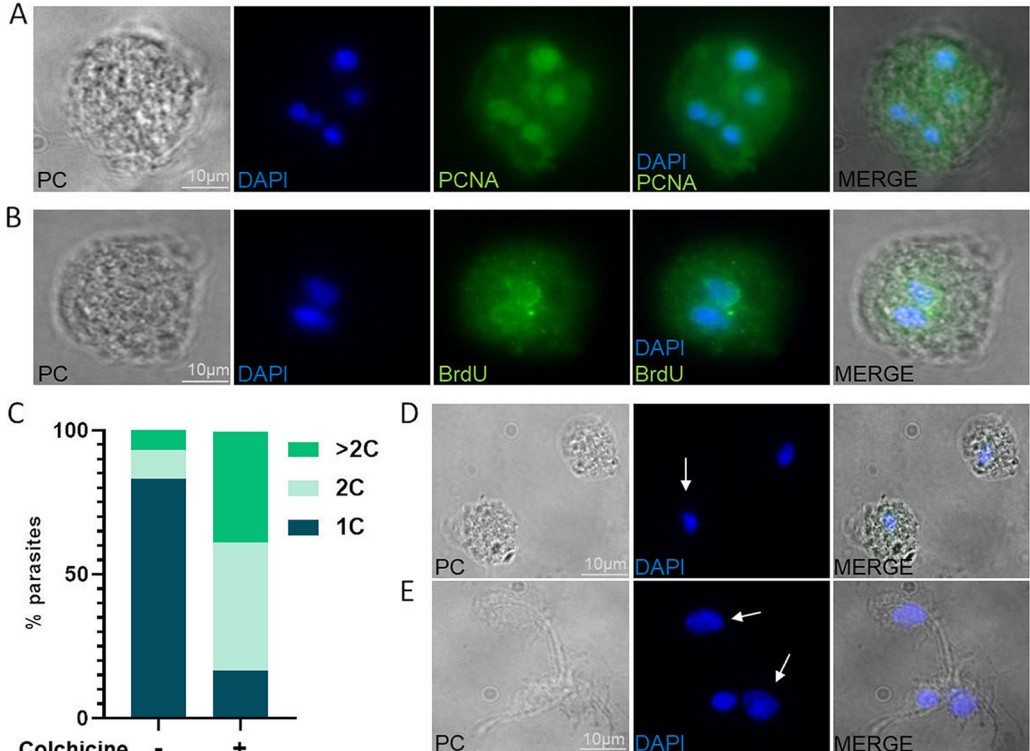

**FIG 9** The genome reduplication process in *T. foetus*. (A) Indirect immunofluorescence image showing PCNA-positive nuclei (green) of *T. foetus* parasites that were grown in the same medium for 36 h. DAPI (blue). PC, phase-contrast image. Scale bar, 10 $\mu$m. (B) Indirect immunofluorescence image of BrdU-positive nuclei (green) of *T. foetus* K parasites that had been serum-starved for 12 h before being incubated with BrdU for 2 h. DAPI (blue). PC, phase-contrast image. Scale bar, 10 $\mu$m. (C) A graph representing the DNA content profile of *T. foetus* K treated with colchicine (and the parasites not treated as a control). 1C, 2C, or >2C, DNA content. (D to E) Representative image of *T. foetus* trophozoites grown in culture media with 1% DMSO (control), and *T. foetus* grown in culture media with colchicine for 18 h. The white arrows indicate the nuclei. Scale bar, 10 $\mu$m.

1, 2, or more than 2 nuclei per parasite in standard growth media conditions. In this sense, several previous studies have documented the existence of multinucleate forms in the life-history of trichomonads, but relevance of this phenomenon has not been analyzed in detail. There are reports that mention the presence of very large individuals with more than 2 nuclei in *T. foetus* (48, 49), *T. vaginalis* (50, 51), *Trichomonas lacerate* (25), *Gigantomonas herculean* (52), as well as the existence of *Trichomastix serpentis* giant forms that could originate 3 or 4 daughter cells (25). Moreover, the occurrence of multiple mitosis in fresh material isolated from natural hosts in *Trichomonas muris*, *Trichomonas augusta*, *Tetratrichomonas prowazeki*, and *Eutrichomonas serpentis* has been documented (25). Additionally, the existence of multinucleated forms named "pseudocysts" has been proposed in *Trichomonas vaginalis* (29, 53), *Tritrichomonas muris* (54), and *Trichomonas tenax* (55). The pseudocysts formation has been described to be induced by temperature variations, drugs, and iron depletion in the *T. vaginalis* media (56, 57); or by temperature variation, iron depletion, and chemicals in *T. foetus* (58, 59).

Multinuclearity has been described as a highly relevant process in different organisms. The emergence of multinucleated cells in plants has been linked to an increase in cell size and, as a result, the organism's overall growth (60). In ameba species, multinucleate cells appear to be beneficial for growth population (61). Under axenic growth conditions, trophozoites of *Entamoeba histolytica* contain heterogenous amounts of DNA due to the presence of both multiple nuclei, and different amounts of DNA in individual nuclei; and the formation of multinucleated cells in this organism is favored by the absence of nutrients (62). In *Acanthamoeba castellanii*, multinuclearity

represents a physiological adaptation of non-adherent parasites that represents an advantage for posterior proliferation (61). *Plasmodium* parasites use multinucleation in very different adaptive contexts. Throughout their life cycle, these parasites regularly go through polyploid and multinucleated phases (63). Additionally, the regulation of progeny number could also be influenced by extrinsic host factors, and the parasite adapt its multiplication depending on the presence of certain nutrients or stress factors (64–66). Here, we demonstrated that multinucleated parasites are present in standard growth conditions, as well as under nutrient starvation conditions in *T. foetus* and *T. vaginalis*. Interestingly, only a low percentage of these multinucleated parasites were CFW positive. Moreover, the daughter nuclei of these multinucleated cells may not remain synchronous during the karyokinesis process, since we observed parasites with 3, 5, and 7 nuclei. Finally, we also revealed that nutrient depletion generates monoclucleated polyploidie parasites, of which only a small percentage are CFW positive. In summary, these results propose the existence of novel dormant forms of resistance (multinucleated or mononucleated polyploid parasites), different than the previously described pseudocysts, that are formed under stress conditions. In this context, we also consider that the change in shape of the parasite, from pyriform to spherical, could be due to osmotic shock, which should be considered in future analyses.

The multiple fission is defined as a division mode that produce more than 2 daughter cells per division by peripheral budding from the plasma membrane of a polyploidy or multinucleated mother cell (25). Some algae and some protozoans regularly divide by multiple fissions (39), where the nucleus undergoes several mitotic divisions. After the nuclear divisions are complete, the cytoplasm separates to form individual cells. In this sense, some authors have described multiple fission as a normal phase of widespread occurrence in the life cycle of *Trichomonas augusta* and other trichomonads (25). It's known that the cell division planes are dictated by multiple mechanisms to ensure successful cytokinesis, thus, the analysis and interpretation of the plane of division in all organisms is highly significant (67, 68). In this context, it is widely accepted that *T. vaginalis* and *T. foetus* divide by longitudinal binary fission. However, the frequent observation of multinucleated cells with heterogeneity in sites of cytokinesis lead us to consider multiple fission as another possible mechanism of cell division in these parasites. Our results also suggest that such multiple fission in these parasites occurs in different planes: longitudinal, transverse, or oblique. According to this, it has been reported that the direction of the plane of division in binary fission in *Tetratrichomonas gallinarum* may occur in any one of 3 planes: longitudinal, transverse, or oblique (25).

A mechanism that could account for multinucleated cell formation is DNA endoreplication. During the endoreplication, the cells undergo DNA replication in the absence of subsequent cell division (69). This process can enclose different alternatives, such as endocycles and endomitosis. Without cell or nucleus division, the first one consists of repeated S-G phases of all genetic material without the cell entering mitosis (70). This results in a single nucleus with increased ploidy. In endomitosis, the cells condense the chromosomes but do not dissociate them into daughter cells. Instead, they reenter a similar phase to G1, and the S phase starts again, occurs during anaphase, resulting in a single nucleus, or in telophase, resulting in multiple nuclei within a single cell (71). The endoreplication process is highly conserved in evolution, and is used as a form of growth by multiple cell types during the development of many plant and animal species (72). This is a frequent replication mechanism in arthropods, and it has been extensively studied in *D. melanogaster*, in plants, in fish, in mice, and in humans, where some of these cells endoreplicate in response to injury or infection, while others endoreplicate as part of a developmental program (73). It has also been observed in a variety of lower invertebrates, and it is frequently linked to an increase in cell and body size. Furthermore, endoreplication improves a plant's resistance to environmental stress and resource scarcity (in a high-temperature or water-deficit environment), and some

plants use this replication mode as a survival response to genome damage (70). Endoreplicating cells in animals develop resistance to DNA damage by reducing proapoptotic gene expression (70). Tumor progression is also accompanied by endoreplication and polyploidy, and it is known that this process is associated with therapy resistance (74). Endoreplication cycles in the parasite *Entamoeba histolytica* result in cells with polyploid nuclei or multinucleate cells (62). Here, we demonstrated that multinucleated *T. foetus* parasites formed under nutritional stress conditions were actively synthesizing DNA. These results could be indicating that endoreduplication may occur in these parasites during depletion or deprivation of nutrient conditions as an alternative strategy for polyploidization (compared with the restitution nucleus) that these parasites use for fast and efficient multiplication when the optimal conditions are reestablished. In conclusion, multiple genome contents present in a single nucleus or distributed over multiple nuclei in a single cell could represent a survival and multiplication mechanism in these parasites in the face of unfavorable environmental conditions.

Finally, we consider that trichomonads could, therefore, represent an interesting model system to study endoreplication and polyploidy, their consequences, and potential advantages in stress response, development, and disease.

## MATERIALS AND METHODS

**Parasite cultures.** *T. foetus* strains: K (Embrapa, Rio de Janeiro, Brazil) (75), OLI, 97H, V47, and 62H strains (Argentina); and *T. vaginalis* strains with different levels of adherence to host cells: G3 and RU38 (low), B7268 and CDC1132 or MSA1132 (high) (76). *T. foetus* and *T. vaginalis* strains were cultured in Diamond's Trypticase-yeast extract-maltose (TYM) medium supplemented with 10% horse serum and 10 U/mL penicillin/10 $\mu$g/mL streptomycin (Invitrogen). Parasites were grown at 37°C and passaged daily. The TvTSP6-HA construct was generated and transfected into the *T. vaginalis* CDC1132 strain as previously described (41).

**Parasite growth assay.** The kinetic of growth curves were performed using K, OLI, 97H, and V47 *T. foetus* strains, as well as G3 and B7268 *T. vaginalis* strains. For these experiments, $1 \times 10^5$ trophozoites were inoculated in 8 mL of TYM medium, and incubated at 37°C for 72 h. After inoculation, cell counts were recorded at the following times: 20 h, 25 h, 43 h, 48 h, and 67 h using a Neubauer hemocytometer. Growth rates were determined as the natural logarithm of the change in the density of parasites per milliliter at time t compared with that at time zero (initial inoculum) by the following equation: growth rate = [lnCC (t) − lnCC (0)]/(t − 0), where CC (t) and CC (0) are the parasites counts per milliliter at time t and time zero, respectively, and t is the time of incubation (77). The experiments were performed three times, in triplicate.

**Nuclear staining.** The number of nuclei per cell was determined using DAPI (4', 6-diamidino-2-phenylindole) labeling. Parasites in the absence of host cells were incubated at 37°C on glass coverslips for 4 h, then were fixed and permeabilized in cold methanol for 10 min. The cells were washed three times in PBS, and incubated with a 300 nM DAPI stain solution for 5 min, protected from light. After 3 washes with PBS, the coverslips were mounted onto microscope slides using fluoromount mounting media. The parasites' nuclei were stained with DAPI for the nuclear size analysis. The nuclear area in $\mu$m$^2$ was examined using Image J's threshold tool and the "Analyze Particles" function, and the experiments were carried out in triplicate and three times.

**CWS.** Parasites were pipetted onto a glass slide, air dried, and fixed in methanol for 10 min at room temperature. Slides were washed three times in PBS for 5 min each time, and incubated with 0.01% Calcofluor White (Sigma) in PBS, pH 7.2, for 30 min at 26°C. The slides were then washed three times in PBS, mounted on fluoromount, and observed under a phase-contrast and fluorescence microscope (Zeiss Axio Observer 7 inverted fluorescence microscope).

**Immunolocalization experiments.** Parasites, grown in different conditions, were incubated at 37°C on glass coverslips for 4 h, and then fixed and permeabilized in cold methanol for 10 min. The cells were washed and blocked with 5% fetal bovine serum (FBS) in PBS for 30 min, and incubated overnight with the primary antibody diluted in PBS plus 2% FBS. Primary antibodies used in the different immunolocalization experiments were anti-HA (1:500) (Covance), anti-tubulin (1:500), anti-BrdU (1:300), and anti-PCNA monoclonal antibody (Abcam) (1:400). Coverslips were washed with PBS, and then incubated for 1 h at room temperature (RT) with a 1:5000 dilution of Alexa Fluor conjugated secondary antibody (Molecular Probes). The coverslips were mounted onto microscope slips using ProLong Gold antifade reagent with DAPI or 4', 6'-diamidino-2-phenylindole (Invitrogen). All observations were performed on a Zeiss Axio Observer 7 (Zeiss) inverted fluorescence microscope. For image processing, Adobe Photoshop (Adobe Systems) was used.

**Flow cytometry.** To determine DNA content and cell size, parasites (5x10$^6$) were harvested and fixed in 5 mL of ice-cold 100% EtOH at 4°C overnight. Following that, each sample was washed in 1 mL of PBS containing 2% vol/vol horse serum (HS), resuspended in 1 mL of PBS,180 $\mu$g/mL RNase A to digest RNA and 2% vol/vol HS, and incubated for 30 min at 37°C. Then, samples were stained with a 25 $\mu$g/mL propidium iodide (PI) solution, and incubated for 30 min at 37°C prior to flow cytometer analysis. Flow cytometry analysis was carried out using a FACS Calibur flow cytometer (Becton, Dickinson) equipped

with a dual laser system (15 mW 488 nm argon ion laser and a 635 nm red diode laser). For measurement of DNA content, cells were excited with 480 nm light, and emission was measured through 585/42 (for PI fluorescence; FL2). Data from 20,000 cells was recorded, and these sets were analyzed using the FlowJo 7.6 software. Correlation between the light scattering properties of cells with their DNA content was performed by setting electronic gates on the forward scatter (FSC) and side scatter (SSC) profiles, and checking for DNA content of every gate.

**Depletion and deprivation of nutrients.** For depletion of nutrients, *T. foetus* parasites (K and 62H strains) were synchronized by serum starvation for 12 h to increase the abundance of uninucleate parasites in culture media. Then, the parasites were grown in the presence of serum (without media change), and samples were taken every 12 h until their death phase. Samples were fixed, and the nuclei were stained with DAPI and quantified by fluorescence microscopy. To evaluate the DNA content in low amounts or depletion of nutrients, we analyzed the *T. foetus* K and 62H strains in the following culture conditions: UP as a control, SP for 12 h, parasites grown for 36 h without change media (HS+), and parasites grown in the absence of serum for 36 h (HS−). The DNA content was determined by flow cytometry.

For the deprivation of nutrients assay, the parasites were incubated in water for 24 and 48 h at 37°C. Afterwards, these samples were fixed, and the nuclei were stained with DAPI and quantified by fluorescence microscopy. The DNA content was determined by flow cytometry.

**Analysis of parasites' final cell division.** To evaluate reversibility of multinucleated parasites induced by nutrient depletion, parasites were grown in complete culture medium, as well as in serum-free medium for 36 h, and then recovered in fresh medium in the presence of serum. Then, the parasites were counted in a Neubauer hemocytometer at 4 h after medium change, and the growth rates were calculated.

To analyze whether the formation of spherical structures formed during nutrient deprivation is a reversible process, parasites cultured for 24 h in water were centrifuged and grown in standard culture medium for 24, 48, and 72 h at 37°C. The parasites were counted in a Neubauer hemocytometer, and the percentages of spherical parasites and trophozoites were calculated.

**BrdU incorporation.** *T. foetus* serum-starved parasites (12 h) were incubated with BrdU for 2 h. The parasites were fixed for 15 min in 4% paraformaldehyde at RT. The cells were washed with PBS, and incubated with 2 M HCl for 20 min at RT. Then, we add 0.1 M sodium borate pH 8.5 for 2 min at RT. The parasites were washed with PBS, and permeabilized with 0.2% Triton X-100 and 3% BSA in PBS for 5 min, washed three times with PBS/BSA for 10 min each, and incubated with anti-BrdU antibody diluted in PBS/2% FBS. Then, the parasites were washed three times with PBS, and incubated for 1 h with the secondary antibody. Finally, we washed the parasites three times, for 10 min each, with PBS, and added DAPI to stain the DNA.

**Colchicine treatment.** The parasites were incubated in TYM medium with colchicine (1.5 mM) dissolved in dimethyl sulfoxide (DMSO). The cells were maintained under drug pressure for 18 h at 37°C. All reagents were purchased from Sigma. Control experiments were performed using trophozoites grown with 1% DMSO. Afterwards, we determined the relative cellular DNA content. Parasites were fixed in 5 mL of ice-cold 100% EtOH, and incubated at 4°C overnight. Following that, each sample was washed in 1 mL of PBS containing 2% vol/vol HS, resuspended in 1 mL of PBS, 180 g/mL RNase A, and 2% vol/vol HS, and incubated for 30 min at 37°C. Then, samples were stained with a 25 $\mu$g/mL propidium iodide (PI) solution, and incubated for 30 min at 37°C prior to flow cytometer analysis. Also, the nuclei were stained with DAPI and observed by a fluorescence microscope.

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
