## [Reviewer comments · Microbiology Spectrum]

Microbiology Spectrum

***Tritrichomonas foetus* cell division involves DNA endoreplication and multiple fissions**

Veronica Coceres, Lucrecia Iriarte, Cristian Martínez, and Natalia de Miguel

Corresponding Author(s): Veronica Coceres, Instituto Tecnológico de Chascomus

Review Timeline:

Submission Date:	August 26, 2022
Editorial Decision:	November 16, 2022
Revision Received:	December 1, 2022
Editorial Decision:	January 9, 2023
Revision Received:	January 13, 2023
Accepted:	January 16, 2023

Editor: guido favia

Reviewer(s): The reviewers have opted to remain anonymous.

Transaction Report:

DOI: <https://doi.org/10.1128/spectrum.03251-22>

November 16, 2022

Dr. Veronica Mabel Coceres
Instituto Tecnológico Chascomús (INTECH)
Laboratorio de Parásitos Anaerobios
Intendente Marino km 8.2
Chascomús, Buenos Aires 7130
Argentina

Re: Spectrum03251-22 (*Tritrichomonas foetus* cell division involves DNA endoreplication and multiple fissions)

Dear Dr. Veronica Mabel Coceres:

Link Not Available

Sincerely,

guido favia

Journals Department
Reviewer comments:

Reviewer #1 (Comments for the Author):

This is an interesting paper that investigates a currently poorly studied feature of the biology of trichomonads, endoreplication and multinucleated forms. As trichomonads are common parasites of humans and animals it is of great interest to gain basic knowledge on their molecular cell biology. Several issues have been identified that should be considered by the authors to improve this interesting manuscript.

Main comments

The title also does not reflect the fact that the presented data cover two species of trichomonads.

The introduction should differential multinucleated cell derived by nuclear divisions, a coenocyte, versus these derived by cell fusion, a syncytium, and cite one or more appropriate review(s).

There should also be a more detailed introduction on the phylogenetic relationship between *Trichomonas* and *Tritrichomonas* species. To what extent are they closely/distantly related? This is important to provide an appropriate comparative perspective for the data presented in the manuscript.

In the Experimental procedures section

Parasites cultures: When listing strains their origins should be described and where relevant citations provided. Also, anything known about their virulence, from either in vivo or in vitro perspectives.

Similarly, what is the origin of the plasmid TvEpNeo/TvTSP6-HA?

Figure 1: The description of panel C needs to be made more informative and precise. What does standard error refer to? Standard deviation (SD) or standard error of the mean (SEM)? These error bars should be SD as these are data from triplicates only. The time frame of the experiment needs to be described in the legend; the measured growth rate was over what period?

Figure 2: For panel C a more discriminative combination of colours is required to help the reader to better differentiate the C values.

Figure 3: Same with panel C for that figure. Line 319: the number listed are probably nm rather than μm ? Same with panel D. All listed units need to be carefully checked throughout the manuscript.

The sentence, lines 337-9 needs to be made more precise, what does "a cytoskeleton" exactly refer too? This should be made explicit throughout the manuscript, including the preceding sentence and the legends of Figure 4 and 5.

Figure 8 and associated figure S1. The legend of this figure needs to be more detailed and precise. Also, I could not find any file with Fig. S1?

Lines 442-3: all relevant data should be shown, either in a main figure or supplementary figure.

Additional information is required relating to the anti-PCNA antibody? Including its source and the nature or the antigen. Any relevant publication should be cited too.

Lines 505-8. This sentence needs to be made more precise/clearer, not sure what the authors are referring too in the current version. For instance, why would you compare bacteria and eukaryotes in the context of this discussion on trichomonads that are eukaryotes?

Additional comments

The statement lines 63-66 should be supported by a reference.

Line 73: "... till to date (20, 21)" the "to" is currently missing.

The sentence line 77-78 is difficult to understand, should be rewritten.

Line 263: The use of "normal" seems an inappropriate term in relation to the presented data in this manuscript. Indeed, the content of DNA is clearly heterogenous and is dependent on the environmental conditions. This sentence should be made more precise.

Reviewer #2 (Comments for the Author):

The manuscript entitled "*Tritrichomonas foetus* cell division involves DNA endoreplication and multiple fissions" by Iriarte et al. investigates the cell division process of a protozoan parasite involved in venereal diseases in bovine. In particular, the authors analysed the phenomenon of multinuclearity, which occurs in protozoa, but also in insects, plants, fungi and higher organisms. This topic is relevant to understand the mechanism of "genome plasticity", which has significant consequences from the genetic point of view (in terms of variable genome content), from the cell biology perspective (in terms of cell division process or cytokinesis) and in tight connection with the cellular stress response, just to mention three relevant issues. Multinuclearity, production of enucleated daughter cells upon cell division, and increase of cellular DNA content are rather well known processes, with extensive literature data available. The authors, in this manuscript, provide novel and original findings regarding *T. foetus*, with a focus on the relationship between nutrient starvation and endoreplication of nuclear DNA .

The manuscript is well written, the experimental data described are scientifically sound and well performed. Each technique

employed in this work is critically described and discussed.

Before considering for publication, however, there are few points that should be carefully addressed:

1. (line 23, line 105) "abundant cells" should be rephrased
2. (line 148) Authors should provide the full name of the acronym FBS
3. (lines 374-376) the sentence should be rephrased
4. (line 430) the word "contain" is present twice in the same sentence, leading to an odd meaning
5. (experiment shown in Fig.7) 24 hours incubation in water induces a change in the shape of the parasite, from pyriform to spherical parasites with internalized flagella. The authors should provide more detailed information related to this phenomenon:
5a) if it has ever been observed with closely related parasites;
5b) if the interpretation is restricted to the lack of nutrients, or if the osmotic shock may play a role in this phenotype.
6. the authors should provide (tentative) suggestions/hypothesis to explain the behavior of T.foetus K strain, which shows the highest degree of multinuclearity among the five strains compared in this study.

Staff Comments:

Preparing Revision Guidelines

Please return the manuscript within 60 days; if you cannot complete the modification within this time period, please contact me. If you do not wish to modify the manuscript and prefer to submit it to another journal, please notify me of your decision immediately so that the manuscript may be formally withdrawn from consideration by Microbiology Spectrum.

Reviewer comments

Initially, we would like to thank the Reviewers and Editor for their valuable time and comments which definitely contributed to improve the manuscript. We have appreciated their compliments, suggestions, and criticism on the quality, strengths and limitations of our main findings and writing. We would also like to thank the opportunity to answer the questions and concerns about our manuscript. We have accepted the points raised by the referees and the alterations are highlighted in blue in the revised manuscript. Thank you for your efforts on our behalf, and we hope to hear from you at your earliest convenience.

Reviewer #1:

This is an interesting paper that investigates a currently poorly studied feature of the biology of trichomonads, endoreplication and multinucleated forms. As trichomonads are common parasites of humans and animals it is of great interest to gain basic knowledge on their molecular cell biology. Several issues have been identified that should be considered by the authors to improve this interesting manuscript.

Main comments

The title also does not reflect the fact that the presented data cover two species of trichomonads.

Response: We truly appreciate your comments, and thank you for the opportunity to clarify your query. We did not include both parasites in the title because we only analyzed the *T. foetus* endoreplication in detail.

The introduction should differentiate multinucleated cell derived by nuclear divisions, a coenocyte, versus those derived by cell fusion, a syncytium, and cite one or more appropriate review(s).

Response: We really thank for this reviewer suggestion; now, we have modified the introduction.

There should also be a more detailed introduction on the phylogenetic relationship between *Trichomonas* and *Tritrichomonas* species. To what extent are they closely/distantly related? This is important to provide an appropriate comparative perspective for the data presented in the manuscript.

Response: We have included in the introduction details about the phylogenetic relationship between both parasites

In the Experimental procedures section

Parasites cultures: When listing strains their origins should be described and where relevant citations provided. Also, anything known about their virulence, from either in vivo or in vitro perspectives.

Response: We really appreciate the reviewer's correction. We have modified the text accordingly.

Similarly, what is the origin of the plasmid TvEpNeo/TvTSP6-HA?

Response: Thanks for this comment. We have included the plasmids' information in the manuscript.

Figure 1: The description of panel C needs to be made more informative and precise. What does standard error refer to? Standard deviation (SD) or standard error of the mean (SEM)? These error bars should be SD as these are data from triplicates only. The time frame of the experiment needs to be described in the legend; the measured growth rate was over what period?

Response: We really appreciate the reviewer's correction. We have modified the text accordingly.

Figure 2: For panel C a more discriminative combination of colours is required to help the reader to better differentiate the C values.

Response: We have followed the reviewer's suggestion and modified the figure accordingly.

Figure 3: Same with panel C for that figure.

Response: We have modified the figure.

Line 319: the number listed are probably nm rather than μm ? Same with panel D. All listed units need to be carefully checked throughout the manuscript.

Response: We truly appreciate the reviewer for pointing this out. Now, we have changed the manuscript and figures accordingly.

The sentence, lines 337-9 needs to be made more precise, what does "a cytoskeleton" exactly refer to?. This should be made explicit throughout the manuscript, including the preceding sentence and the legends of Figure 4 and 5.

Response: This is an important point raised by the Reviewer, and we truly appreciate that. We have modified the text based on your comments.

Figure 8 and associated figure S1. The legend of this figure needs to be more detailed and precise. Also, I could not find any file with Fig. S1?

Response: We really appreciate the reviewer's correction. We have modified the legend and the figures.

Lines 442-3: all relevant data should be shown, either in a main figure or supplementary figure.

Response: We added the figure as Fig. 9D.

Additional information is required relating to the anti-PCNA antibody? Including its source and the nature or the antigen. Any relevant publication should be cited too.

Response: We are really grateful for this reviewer's suggestion, and we have now added the information about the anti-PCNA antibody.

Lines 505-8. This sentence needs to be made more precise/clearer, not sure what the authors are referring to in the current version. For instance, why would you compare bacteria and eukaryotes in the context of this discussion on trichomonads that are eukaryotes?

Response: We really thank the reviewer for this correction; now, we have modified the manuscript.

Additional comments

The statement lines 63-66 should be supported by a reference.

Response: We have now added the references.

Line 73: "... till to date (20, 21)" the "to" is currently missing.

Response: We really thank this reviewer for this correction, and we have modified the text.

The sentence line 77-78 is difficult to understand, should be rewritten.

Response: We have modified the text accordingly.

Line 263: The use of "normal" seems an inappropriate term in relation to the presented data in this manuscript. Indeed, the content of DNA is clearly heterogenous and is dependent on the environmental conditions. This sentence should be made more precise.

Response: Thanks for this comment. We have modified the text accordingly.

Reviewer #2 (Comments for the Author):

The manuscript entitled "Tritrichomonas foetus cell division involves DNA endoreplication and multiple fissions" by Iriarte et al. investigates the cell division process of a protozoan parasite involved in venereal diseases in bovine. In particular, the authors analysed the phenomenon of multinuclearity, which occurs in protozoa, but also in insects, plants, fungi and higher organisms. This topic is relevant to understand the mechanism of "genome plasticity", which has significant consequences from the genetic point of view (in terms of variable genome content), from the cell biology perspective (in terms of cell division process or cytokinesis) and in tight connection with the cellular stress response, just to mention three relevant issues. Multinuclearity, production of enucleated daughter cells upon cell division, and increase of cellular DNA content are rather well known processes, with extensive literature data available. The authors, in this manuscript, provide novel and original findings regarding T. foetus, with a focus on the relationship between nutrient starvation and endoreplication of nuclear DNA.

The manuscript is well written, the experimental data described are scientifically sound and well performed. Each technique employed in this work is critically described and discussed.

Before considering for publication, however, there are few points that should be carefully addressed:

(line 23, line 105) "abundant cells" should be rephrased

Response: We really thank this reviewer for this correction, and we have modified the text.

(line 148) Authors should provide the full name of the acronym FBS

Response: We have now added the full name of "FSB".

(lines 374-376) the sentence should be rephrased

Response: Thanks for this suggestion. We have rephrased the text accordingly.

(line 430 or 239?) the word "contain" is present twice in the same sentence, leading to an odd meaning

Response: Thanks for this comment. We have rephrased the text

(experiment shown in Fig.7) 24 hours incubation in water induces a change in the shape of the parasite, from pyriform to spherical parasites with internalized flagella. The authors should provide more detailed information related to this phenomenon:

5a) if it has ever been observed with closely related parasites;

5b) if the interpretation is restricted to the lack of nutrients, or if the osmotic shock may play a role in this phenotype.

Response: This is an important and interesting point raised by the Reviewer. We have modified the text accordingly.

the authors should provide (tentative) suggestions/hypothesis to explain the behavior of T.foetus K strain, which shows the highest degree of multinuclearity among the five strains compared in this study.

Response: It has been reported that there is phenotypic heterogeneity among *Trichomonas vaginalis* isolates and *Tritrichomonas foetus* isolates, and here, we hypothesize that multinuclearity of the K strain could be associated with this phenotypic heterogeneity (<https://doi.org/10.1128/iai.53.2.285-293.1986>; <https://doi.org/10.1016/j.exppara.2012.03.015>; <https://doi.org/10.1007/s00436-004-1251-0>).

January 9, 2023

Dr. Veronica Mabel Coceres
Instituto Tecnológico de Chascomus
Laboratorio de Parásitos Anaerobios
Intendente Marino km 8.2
Chascomús, Buenos Aires 7130
Argentina

Re: Spectrum03251-22R1 (*Tritrichomonas foetus* cell division involves DNA endoreplication and multiple fissions)

Dear Dr. Veronica Mabel Coceres:

Thank you for submitting your manuscript to Microbiology Spectrum. We invite you to just address the two minor modifications indicated by reviewer 2.

Link Not Available

Sincerely,

guido favia

Journals Department
Reviewer comments:

Reviewer #1 (Comments for the Author):

All key issues identified in the original version of the manuscript have been addressed. Just two details to consider: Scale bars should be added to the cell imaging figures (apologies I missed that detail). Are the surface area of the nuclei not 10X smaller than indicated in the main text and Fig. 2?

Reviewer #2 (Comments for the Author):

Before considering for publication the manuscript entitled "*Tritrichomonas foetus* cell division involves DNA endoreplication and

multiple fissions" by Iriarte et al. two points should be carefully addressed:

1. (line 467) the authors provide a list of several *T. foetus* strains. In the next sentence they mention: "Both trichomonads were cultured...". The sentence should be modified to indicate which strain (two strains?) was grown in TYM supplemented with 10% horse serum and which strain was supplemented with 10U/ml penicillin and/or (?) streptomycin;
2. (line 343) as already suggested in the previous list of comments, the word "contain" is present twice in the same sentence, leading to an odd meaning.

Staff Comments:

Preparing Revision Guidelines

Please return the manuscript within 60 days; if you cannot complete the modification within this time period, please contact me. If you do not wish to modify the manuscript and prefer to submit it to another journal, please notify me of your decision immediately so that the manuscript may be formally withdrawn from consideration by Microbiology Spectrum.

Before considering for publication the manuscript entitled "*Tritrichomonas foetus* cell division involves DNA endoreplication and multiple fissions" by Iriarte et al. two points should be carefully addressed:

1. (line 467) the authors provide a list of *T. foetus* strains. In the next sentence they mention: "Both trichomonads were cultured...". The sentence should be modified to indicate which strain was grown in TYM supplemented with 10% horse serum and which strain was supplemented with 10U/ml penicillin and/or (?) streptomycin;
2. (line 343) as already suggested in the previous list of comments, the word "contain" is present twice in the same sentence, leading to an odd meaning.

Reviewer comments

We really appreciate the comments which definitely contributed to improve the manuscript. We have accepted the points raised by the referees and the alterations are highlighted in blue in the revised manuscript.

Reviewer #1:

All key issues identified in the original version of the manuscript have been addressed. Just two details to consider:

Scale bars should be added to the cell imaging figures (apologies I missed that detail).

Response: We truly appreciate the reviewer for pointing this out. Now, we have added the scale bars to the figures.

Are the surface area of the nuclei not 10X smaller than indicated in the main text and Fig. 2?

Response: Thanks for this comment. In general, it has been reported that *T. foetus* is variable in size, measuring 5.7 to 13.0 μm \times 3.9 to 5.5 μm and *T. vaginalis* varies in size and shape, with a length of 10-20 μm and width of 7 μm . According to this, the nuclei are variable in size, and our results about the nuclei's surface area in μm^2 are consistent with reports that analyze the length and width of different trichomonads (including *T. foetus* and *T. vaginalis*) (<https://doi.org/10.1016/j.molbiopara.2008.06.004>).

Reviewer #2:

Before considering for publication the manuscript entitled "Trichomonas foetus cell division involves DNA endoreplication and multiple fissions" by Iriarte et al. two points should be carefully addressed:

1. (line 467) the authors provide a list of several *T. foetus* strains. In the next sentence they mention: "Both trichomonads were cultured...". The sentence should be modified to indicate which strain (two strains?) was grown in TYM supplemented with 10% horse serum and which strain was supplemented with 10U/ml penicillin and/or (?) streptomycin;

Response: We really appreciate the reviewer's correction. We have modified the text accordingly.

2. (line 343) as already suggested in the previous list of comments, the word "contain" is present twice in the same sentence, leading to an odd meaning.

Response: Thanks for this comment. In the last version, we couldn't find the word contain twice in the same sentence in line 343. Anyway, we have revised the manuscript, and we checked that this word does not appear twice in any other sentence.

the authors should provide (tentative) suggestions/hypothesis to explain the behavior of T.foetus K strain, which shows the highest degree of multinuclearity among the five strains compared in this study.

Response: It has been reported that there is phenotypic heterogeneity among *Trichomonas vaginalis* isolates and *Tritrichomonas foetus* isolates, and here, we hypothesize that multinuclearity of the K strain could be associated with this phenotypic heterogeneity (<https://doi.org/10.1128/iai.53.2.285-293.1986>; <https://doi.org/10.1016/j.exppara.2012.03.015>; <https://doi.org/10.1007/s00436-004-1251-0>).

January 16, 2023

Dr. Veronica Mabel Coceres
Instituto Tecnológico de Chascomus
Laboratorio de Parásitos Anaerobios
Intendente Marino km 8.2
Chascomús, Buenos Aires 7130
Argentina

Re: Spectrum03251-22R2 (*Trichomonas foetus* cell division involves DNA endoreplication and multiple fissions)

Dear Dr. Veronica Mabel Coceres:

Your manuscript has been accepted, and I am forwarding it to the ASM Journals Department for publication. You will be notified when your proofs are ready to be viewed.

Sincerely,

guido favia
Editor, Microbiology Spectrum
